# Serum Calprotectin and Chemerin Concentrations as Markers of Low-Grade Inflammation in Prepubertal Children with Obesity

**DOI:** 10.3390/ijerph17207575

**Published:** 2020-10-18

**Authors:** Grażyna Rowicka, Hanna Dyląg, Magdalena Chełchowska, Halina Weker, Jadwiga Ambroszkiewicz

**Affiliations:** 1Department of Nutrition, Institute of Mother and Child, 01-211 Warsaw, Poland; hanna.dylag@imid.med.pl (H.D.); halina.weker@imid.med.pl (H.W.); 2Department of Screening Tests and Metabolic Diagnostics, Institute of Mother and Child, 01-211 Warsaw, Poland; magdalena.chelchowska@imid.med.pl (M.C.); jadwiga.ambroszkiewicz@imid.med.pl (J.A.)

**Keywords:** calprotectin, chemerin, obesity, chronic low-grade inflammation, prepubertal children

## Abstract

In adults, obesity is associated with chronic low-grade inflammation, which may cause long-term adverse health consequences. We evaluated whether obesity in prepubertal children also generates this kind of inflammation and whether calprotectin and chemerin may be useful markers for early detection of such inflammation in this group of children. The study population included 83 children aged 2 to 10 years; 62 with obesity and without components of metabolic syndrome and 21 healthy controls with normal body weight. White blood cell (WBC) count, concentrations of C-reactive protein (CRP), interleukin-6 (IL-6), calprotectin, and chemerin were determined in peripheral blood. Our study showed that in the group with obesity, serum concentrations of calprotectin and chemerin, as well as CRP were significantly higher as compared with the controls. We found a significant positive correlation between serum chemerin concentrations and BMI z-score (r = 0.33, *p* < 0.01) in children with obesity. Chemerin concentration was also positively correlated with CRP level (r = 0.36, *p* < 0.01) in the whole group of children. These findings suggest that obesity may generate chronic low-grade inflammation as early as in the prepubertal period which can be indicated by significantly higher serum concentrations of calprotectin and chemerin. Calprotectin and especially chemerin seem to be promising indicators of this type of inflammation in children with obesity, but the correlation between these markers requires further research.

## 1. Introduction

In recent decades, the worldwide prevalence of overweight and obesity has exhibited a steady growth among adults and also in children [1,2]. This is worrying because it has been demonstrated that excess body weight in childhood is associated with a greater risk of developing overweight as well as obesity and its complications in adulthood [3,4].

It has been proven that obesity in adults and adolescents can be accompanied by chronic inflammation, the pathophysiology of which seems to be complex [5]. Excessive hyperplasia or hypertrophy of fat cells may lead to cellular damage by inducing adipose tissue hypoxia. Hypoxia is also proposed to activate the innate immune system, which leads to overproduction of proinflammatory mediating cytokines, adipokines, and chemokines [6]. The released proinflammatory factors are also responsible for the overproduction of reactive oxygen species (ROS), which can cause oxidative stress and free radical damage of important cell structures [7]. All these factors cause an inflammation in adipose tissue that propagates the systemic inflammation associated with the development of obesity-related comorbidities [8,9,10].

Recently, calprotectin and chemerin are noted to be as important factors involved in the inflammatory process cascade. Calprotectin is a protein complex which occurs mainly in the cytoplasm of neutrophils, but also in monocytes and macrophages, playing an important role in the leukocyte chemotaxis, adhesion, and phagocytosis. Moreover, the possibility of reversible binding of unsaturated fatty acids (especially arachidonic acid) by calprotectin, probably allows for their use as a substrate for the production of leukotrienes and other inflammatory response mediators. The role of calprotectin in the inflammatory process is also connected with its influence on acquired immunity [11,12].

It has been proven that the calprotectin blood level correlates with the degree of adiposity in both healthy children and children with excess body weight [13,14]. It has also been shown that calprotectin may be more useful for both detecting and monitoring inflammation in adults, related to metabolic complications of obesity, as compared with conventional inflammatory markers, such as white blood cell (WBC) count or C-reactive protein (CRP) [14,15].

Chemerin is an adipokine mainly produced in adipose tissue. This adipokine acts on both paracrine and autocrine functions of adipocytes, stimulating their maturation and differentiation processes, and it also influences angiogenesis in adipose tissue, regulates the innate and acquired inflammatory response, and modulates the expression of genes involved in glucose and lipid homeostasis [16,17]. The pleiotropic properties of chemerin may potentially have an impact on the pathophysiology of excess body weight, and on the predisposition for developing metabolic syndrome [17,18,19].

Although numerous studies have documented the usefulness of calprotectin and chemerin in detecting inflammation, the correlations of those markers in obesity have not been studied [11,15,16,17].

Thus, in the present study, we aimed to determine whether obesity in prepubertal children generates chronic low-grade inflammation and whether calprotectin and chemerin may be useful markers for early detection of such inflammation in this group of children.

## 2. Materials and Methods

### 2.1. Participants

The study protocol was approved by the Ethical Committee of the Institute of Mother and Child (Decision No. 23/2014) in accordance with Declaration of Helsinki principles. Parents of all examined children were informed of the study objectives and gave their informed written consent. 

Between January 2014 and June 2016, we recruited eighty-three children aged 2–10 years, sixty-two obese children, and twenty-one healthy normal body weight children as controls. They were referred to the Gastroenterology Outpatient Clinic at the Institute of Mother and Child in Warsaw with the aim of weight normalization or for health checkups including the assessment of their diets. The criteria for inclusion in the study group were the following: body mass index (BMI) z-score ≥3 SD in children up to 5 years old and BMI z-score ≥2 SD in children over 5 years old as the diagnostic criterion for obesity [20]; prepubertal age (first stage of puberty according to Tanner criteria) [21]; lack of metabolic complications of obesity, as established based on physical examination (including blood pressure measurement); and the results of laboratory tests such as the concentration of glucose, insulin, transaminases, creatinine, and the serum lipid profile. The criteria for inclusion in the control group were the following: body weight BMI z-score of <−1 and +1 >SD, prepubertal age, lack of abnormalities in the physical examination, and correct results of laboratory tests such as in the group of obese children.

Exclusion criteria from the study groups included endocrine diseases and other systemic disorders, infections of various etiologies and localizations, as well as the intake of medications and food supplements.

These were the same groups of children in which we had earlier conducted studies pertaining to the risk of oxidative stress [22].

### 2.2. Anthropometric Measurements 

Body weight (kg) and height (m) were measured in children in underwear, using a calibrated set consisting of electronic scales and stadiometer. In all children, the measurements were performed by the same person. Body weight was measured with the accuracy of 0.1 kg and height with the accuracy of 1 mm. The BMI was calculated according to the formula, body weight (kg)/height squared (m^2^). BMI values were compared with BMI norms for age and sex according to the WHO criteria, thus, obtaining a BMI z-score, which was a normalized relative weight indicator independent of age and sex [20,23]. 

### 2.3. Blood Sampling and Biochemical Analysis

Peripheral blood samples (3.0 mL) were taken in the morning hours from fasting patients. Whole blood in EDTA-containing tubes was analyzed immediately to determine white blood cell (WBC) count concentrations using a Pentra 60 Haematology Analyzer (HORRIBA ABX; Montpellier, France). The remaining blood was centrifuged at 2500× *g*, at 4 °C, for 10 min, serum samples were separated and stored at −70 °C until use. 

Serum levels of CRP were measured using immunoturbidimetric assay on the Cobas Integra auto-analyzer (Roche Diagnostics, Basel, Switzerland).

Concentrations of serum chemerin, calprotectin, and interleukin-6 (IL-6) were determined immunoenzymatically (ELISA) using specific antibodies with high affinity to these proteins. The chemerin level was evaluated using the Human Chemerin ELISA (Mediagnost, Reutlingen, Germany). Assay analytical sensitivity was 0.005 pg/mL, and intra- and inter-assay coefficients of variability (CVs) were below 5.16% and 2.17%, respectively. Levels of calprotectin and IL-6 were determined using the CALPROLAB^TM^ Calprotectin (ALP) ELISA kit (CALPRO AS, Lysaker, Norway) and the IL-6 ELISA set (DRG Instruments GmbH, Marburg, Germany), respectively. The detection limits of these methods were below 5.0 ng/mL for calprotectin and 2.0 pg/mL for IL-6. The intra- and inter-assay CVs were found to be less than 5.0% for calprotectin and 4.2% and 4.4% for IL-6, respectively. 

### 2.4. Statistical Analysis

The Shapiro–Wilk test was used to check the normality of the data distribution. Parametric data were described as means and standard deviations (SDs) and were analyzed using Student’s *t* test. Non-parametric data were expressed as medians and interquartile ranges and were analyzed using the Mann–Whitney U test. The Pearson correlation coefficients (*r*) were calculated to evaluate the statistical association between biochemical parameters and the BMI z-score. The significance level was set as 0.05. The statistical analyses of the results were carried out using STATISTICA 12.0 software (StatSoft Inc., Tulsa, OK, USA). 

## 3. Results

Children in obese and normal weight groups were similar in terms of age (median of 7.5 years, range of 6.3–8.8 years vs. median of 6.4 years, range of 5.5–8.6 years, *p* = 0.130, respectively). As expected, the weight, and also the height, of children with obesity were significantly higher than those of children with normal weight (mean ± SD, 40.2 ± 9.3 kg vs. 22.2 ± 6.6 kg, *p* < 0.001; median of 1.30 m and range of 1.21–1.41 m vs. median of 1.16 m and range of 1.03–1.29 m, *p* = 0.001, respectively). In addition, values of BMI (median of 23.5 and range of 21.9–24.6 vs. median of 15.5 and range of 15.2–16.3, respectively) and BMI z-score (median of 3.0 and range of 2.5–3.5 vs. median of 0.03 and range of 0.5–0.7, respectively) were significantly higher (*p* < 0.001) in obese children as compared with the control group. The history of obesity in children with excess body weight, determined based on anthropometric measurements performed during regular checkups in infancy, at twelve months, and second, forth, sixth, and tenth year of age, was 3.5 (range of 2.6–4.5) years.

Differences between serum concentrations of biochemical parameters in both groups of patients included in the study are presented in Table 1.

In the group with obesity, serum calprotectin, chemerin, and CRP levels were significantly higher as compared with the normal body weight children. In 11.3% of the obese patients, the median value of IL-6 was 145.8 ng/mL (range of 98.2–150.2 ng/mL), while for the remaining children in this group and for all children in the control group, the concentrations of this marker were below the detection level. 

We observed that chemerin concentrations were positively associated with CRP levels (r = 0.36, *p* < 0.01) in children from both groups (n = 83) (Figure 1). There were no significant relationships between levels of remaining studied parameters describing inflammation in both studied groups.

We also found a significant positive correlation between BMI z-score and serum chemerin concentrations (r = 0.33, *p* < 0.01) in children with obesity, while the correlation between BMI z-score and serum calprotectin was on the border of statistical significance (Table 2). 

BMI z-score, a normalized relative weight indicator independent of age and sex; CRP, C-reactive protein; WBC, white blood cell; *p* < 0.05 was a significant level.

## 4. Discussion

The search for markers that could be used for identification of low-grade inflammation, called metabolic or subclinical inflammation in prepubertal obese children, seems to be important, taking into account its relationship with metabolic complications well documented in obese adults. 

The present study showed significantly higher serum concentrations of calprotectin, chemerin, and CRP in children with excessive body weight as compared with the concentration of these parameters in healthy weight children. This observation suggests that obesity can cause low-grade inflammation in prepubertal children, and calprotectin and chemerin seem to be its promising indicators.

The relationship between CRP, a protein of acute phase response, and overweight/obesity has been intensively investigated. It has been shown that the CRP value was higher in patients with obesity and also that CRP elevation correlated with body weight gain and adiposity in both adults and adolescents [24,25,26]. According to observations of children aged 8–16, excess weight is associated with increased CRP concentrations and also with an elevated WBC count, which could not be explained by other factors associated with inflammatory state [27].

Our study did not show differences in WBC count between both groups of children, but it showed significantly higher CRP concentrations in children with obesity, nevertheless, within the reference values. It is possible that the levels of inflammatory markers are dependent on excess body weight and also on a history of obesity [28].

Studies in adults have shown a strong relationship of certain metabolic diseases, for example, diabetes, with elevated levels of CRP, TNF-α, and leptin [29]. CRP has now emerged as one of the most powerful predictors of inflammatory and lipid markers for evidence of cardiovascular events [30]. There are currently no guidelines associating the CRP level and the cardiovascular risk in children, but adults with CRP >3 mg/L are considered to have this risk increased 1.5- to two-fold [31]. According to Vehapoglu et al. [32], CRP values of 4.11 ± 3.10 mg/L found in obese children aged 2–11 years could point to an increased risk of cardiovascular diseases in the future, according to the adult cut-off point. At present, the relationship between CRP and components of metabolic syndrome in children has not been clearly established [33].

In children, increased calprotectin levels in serum have been reported in various chronic inflammatory conditions, such as inflammatory bowel disease, celiac disease, juvenile idiopathic arthritis, lung diseases, and cancer [34].

Observations in adults have shown that serum calprotectin was significantly increased in obese subjects as compared with non-obese subjects and was associated with insulin resistance and type 2 diabetes [35]. 

The study conducted among children (11.7 ± 4.1 years) showed higher calprotectin concentration in obese and overweight children as compared with normal weight subjects. Increased calprotectin levels were related to pathological fasting blood glucose and insulin resistance, while diastolic pressure and BMI were independent factors for its elevated concentration [14].

Similar to this study, higher calprotectin serum level in obese children as compared with normal weight children was shown in the study conducted by Hasan et al. [36]. The authors showed that calprotectin directly correlated with BMI, BMI percentile, BMI z-score, body fat, fat to muscle ratio, leptin, leptin to adiponectin ratio, CRP, IL-6, and MCP-1, and resistin, while it inversely correlated with adiponectin.

In contrast to previous observations, we have not demonstrated a relationship between calprotectin and the remaining inflammatory markers. It cannot be excluded that the results were influenced by the fact that our study concerned only prepubertal children. The prepubertal age is the period when the immunological system matures as well as the immune response. Another factor which may have been important here was the fact that both the number of leucocytes and the CRP concentration in both groups of children were within the reference limits. 

The chemerin concentration correlates with BMI, as well as adipocyte volume and number. In addition, it has been found that, in adults with obesity, weight loss was accompanied by a significant decrease in circulating serum chemerin levels [37,38,39]. The significantly higher chemerin levels in children with obesity and a positive correlation of chemerin with BMI in the obese group was demonstrated in our study. It is believed that it is mainly inflammation that is responsible for its increased synthesis in adults with obesity and type 2 diabetes, however, in obese patients, the stimulating effect of TNF-α on its synthesis was also shown [40].

In both in vitro and in vivo studies, a positive association with chemerin and proinflammatory cytokine concentrations (e.g., TNF-α and IL-6), and adipokines (e.g., leptin and resistin), as well as CRP, and an inverse correlation with anti-inflammatory cytokine and adiponectin have been demonstrated [41,42].

It is possible that elevated chemerin levels found in obese children could point to early stages of obesity-related diseases, for example, cardiovascular disease or diabetes [43,44,45,46,47].

The chemerin correlation with markers of inflammatory vascular processes, for example, endothelial activation intracellular adhesion molecule-1 (ICAM-1) and E-selectin observed in children, could suggest the initial stage of atherogenesis already at this age [43].

Taking into account the observations that point to a relationship between obesity and its complications with subclinical inflammation, it seems that calprotectin and chemerin could be helpful for diagnosis and monitoring of its course, also including evaluating the effectiveness of therapy in children.

A limitation of our study was the relatively small group included in the observation; however, it was similar in number to other research of this kind [45,47].

Another limitation was the fact that the study covered only obese children without components of metabolic syndrome. It would be interesting to compare the concentration of markers that we determined in obese children with and without these components. We are planning to conduct such observations. Other inflammatory markers such as TNF-α, ICAM-1, and E-selectin with a proven relationship with obesity and its complications in adults and adolescents have also not been studied [36,41,42]. In our study, about 90% of children with obesity and in all children with normal body weight, the concentrations of IL-6 were below the detection level. It is possible that the use of a more sensitive method for determining IL-6 would allow for different, more precise results. It cannot be excluded that IL6, which is an acute marker of inflammation is not a good marker for chronic low-grade inflammation which may be associated with obesity in children. 

A strength of our study is the fact that it is one of a few studies pertaining only to prepubertal children. This eliminates the influence of comorbidities seen in adults on the concentrations of the analyzed markers. It may also be important in the context of dependence of chemerin and calprotectin concentrations on age and sex, as found by some authors [14,43]. The regulatory influence of sex hormones on their concentration has been taken into account [14,46,47]. According to our knowledge, it was the first study assessing the correlation of calprotectin with chemerin in this group of children. A positive correlation (regression analysis) has only been observed in adults with psoriasis [48].

Our study aimed to minimize the effect of physical activity and circadian rhythm as potential external confounder [12,19,49,50] on the concentration of inflammatory markers by reducing preanalytical variance (i.e., overnight fasting and morning blood draws). Inflammatory marker concentrations could also show differences depending on the type of material collected for testing. In all children, this material was serum obtained after blood centrifugation, which was stored under the same conditions until determination. 

## 5. Conclusions 

Our study showed that obesity may generate chronic inflammation in children as early as in prepubescence, as indicated by significantly higher serum concentrations of calprotectin, chemerin, and CRP in these children as compared with children of normal body weight. Calprotectin, and especially chemerin, seem to be promising indicators of inflammation associated with obesity in children, but the correlation between these markers requires further research.

## Figures and Tables

**Figure 1 ijerph-17-07575-f001:**
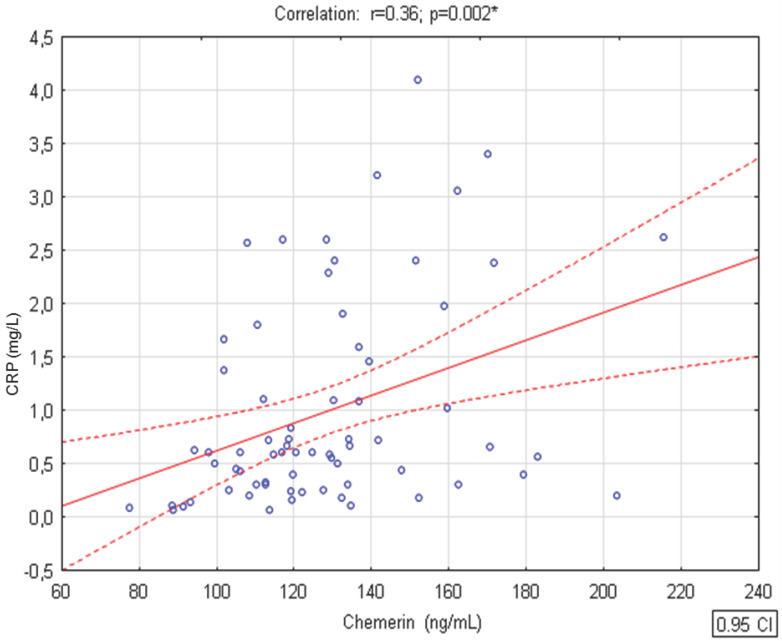
Correlation between serum chemerin and C-reactive protein (CRP) concentrations in the whole group of studied children. r, Pearson’s correlation coefficient

**Table 1 ijerph-17-07575-t001:** Biochemical parameters in children with obesity and normal body weight.

Variable	Children with Obesity (n = 62)	Normal Body Weight Children (n = 21)	*p* Value
Calprotectin (ng/mL) ^+^	976.4 (796.0–1303.6)	797.0 (618.5–1070.7)	0.029
Chemerin (ng/mL) ^+^	130.5 (113.6–149.4)	113.8 (91.4–129.8)	0.006
CRP (mg/L) ^+^	0.66 (0.43–1.66)	0.24 (0.10–0.59)	0.008
WBC count (10^9^/L) ^++^	6.8 (1.6)	7.4 (1.6)	0.150

^+^ Data are presented as medians and interquartile ranges (1Q–3Q) Mann–Whitney U test. ^++^ Data are presented as means and standard deviations (SDs) Student’s *t* test. CRP, C-reactive protein; WBC, white blood cell; *p* < 0.05 was a significant level.

**Table 2 ijerph-17-07575-t002:** Simple correlations between BMI z-score and biochemical parameters in children with obesity.

Variable	BMI z-Score	
	r	*p* Value
Calprotectin (ng/mL)	0.21	0.060
Chemerin (ng/mL)	0.33	0.003
CRP (mg/L)	0.20	0.093
WBC count (10^9^/L)	−0.02	0.882

r, Pearson’s correlation coefficient.

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
