# Peer review of "Serum Calprotectin and Chemerin Concentrations as Markers of Low-Grade Inflammation in Prepubertal Children with Obesity"

_ijerph, 2020, doi:10.3390/ijerph17207575_

Round 1

Reviewer 1 Report

Thank you for the opportunity to review this article. The work is interesting, but some aspects should be taken into account before publication.

Comments and suggestions for Authors:

  • Please standardize the font for affiliation
  • Line 14 - Remove bold with "In"
  • Please correct the references according to the guidelines for authors (font, line spacing)
  • Language needs to be clarified throughout to include person first language

Introduction:

It is incomprehensible for me to present such an extensive introduction without literature review. Authors should include a literature review in the introduction. The type “article” requires to clearly provide rationale for the study (with research questions and hypotheses) and clearly identify what this study adds to the current literature on this topic. Therefore this part should be shortened.

Authors should improve the introduction including the latest articles published for example in the MDPI platform or others, about other research in this field

Methodology:

  • Why authors include exactly 83 children? How many invitations did you send? Please write more about recruitment of the study group.
  • Line 116 - add citation to the WHO criteria
  • Lines: 114-117 - The information about the performed measurements is very short and insufficient. Under what conditions was the measurement performed? how many times? What was the accuracy level for weight, height?

Results:

Tables 1 and 2: Below the table, please add a description of p.

Figure 1: The names of the figures are signed below the figure, not above it.

Discussion:

Authors should improve the discussion including the latest articles about this topic.

Additional limitations of this study should be noted. Strengths of this study should be noted.

Reviewer 2 Report

In their manuscript, entitled   "Serum calprotectin and chemerin concentrations as markers of low grade inflammation in prepubertal children with obesity"
Grażyna Rowicka et al. provide data on the inflammatory state in childhood obesity before the onset of the metabolic syndrome.

The authors determined that obesity significantly induced serum concentrations of calprotectin and chemerin, as well as CRP, with
a significant positive correlation between serum chemerin concentrations and BMI z-score. The work is appropriate for the journal.
  However there are some observations that must be addressed before the manuscript can be accepted:   1. English language editing is required.   2. MAJOR:
It is already well accepted that chemerin plays a role in obesity-related inflammation not only in in adults,
but also in children, and other pediatric studies have shown correlations of chemerin to CRP.
While general information is presented on the inflammatory markers, the following important studies were not introduced, discussed or cited (see below). Thus, the impact of the results presented remains somewhat vague.
Pubmed IDs (PMID):
30030480,
26261457,
30030480,
22438234 (+Vascular changes in children),
28551507 (5-17years + CRP correlation)

3. As all measurements were perfomed standardized under fasting conditions. With respect to metabolic syndrom it would be nice to apply
Massspectrometric metabolomics to left over serum - as the progession to metabolic syndrome might be hard to grasp otherwise..
  4. How do your levels of inflammatory markes compare to levels detected in children with metabolic syndrome?   5. Did you consider measuring chemerin in urine?   6. CRP results have already be published by your group.
Harmful and Beneficial Role of ROS 2017, Volume 2017 |Article ID 5621989 | https://doi.org/10.1155/2017/5621989. Grażyna Rowicka et al.
Total Oxidant and Antioxidant Status in Prepubertal Children with Obesity
- as cited in the methods.
Please ensure that no self-plagiarism occurs.   7. Minor: Please be more precise: a) a relatively short obesity duration --->short history of... (the exact duration or onset are unkown)
b) when talking about "adolescents and older children" what is the difference?

Reviewer 3 Report

The authors investigated the level of IL-6, CRP, chemerin and calprotectin in the blood of children. The authors considered the hypothesis of chronic inflammation in obese patients. In the scientific literature, there are many works devoted to this problem, it has been studied quite well. What is the peculiarity and interest for the scientific community of your results? I propose, among other things, to make comparisons with adults in terms of the studied parameters, for example, there are associations of TNF-a chimer and CRP in adult obese patients and with its complication T2DM. Chemerin is used as a marker of metabolic complications, but it has protective effects against metabolic complications. With a long chronic course of AT inflammation, it decreases and another protein isoform predominates because of this, another receptor is activated. Which leads to the progression of inflammation and metabolic complications. Many works have demonstrated its multidirectional effects depending on anthropometric and biochemical parameters.
There is not enough data for publication at a good level, since they do not reveal any mechanisms, new diagnostic methods, new diagnostic markers, and potential use in therapy. It is necessary to add a specific hypothesis and justification of the importance and relevance of these molecules, otherwise the results of the article are lost against the background of many similar works.
The connection between chemerin and calprotectin is not clear from the introduction, why the molecules associated with them have not been studied?
What inclusion / exclusion criteria were used? Have the children taken any medications? Did the hormonal composition affect the result? Were there sex addictions?
Is there a research permit? Is there a study registration?

Round 2

Reviewer 1 Report

Thank you for the opportunity to review this resubmission.  Authors have done a nice job addressing reviewers' comments. Thank you. I am ok with acceptance.

Reviewer 2 Report

The authors have revised the manuscript following my suggestions. However, there are a few minor issues that still need to be addressed (see below). Most of these reflect English language editing.  

Page1

line 15: whether, the calprotectin and chemerin
>> whether calprotectin and chemerin

line 18-19: The peripheral blood levels of white blood cell (WBC) count and serum concentrations of C-reactive protein (CRP), interleukin-6 (IL-6), calprotectin and chemerin were determined.
>>
 White blood cell (WBC) count […] were determined in peripheral blood.

Line 26: as in prepubertal period >> as in the prepubertal period

Line 33: exhibits >> exhibited

Page 2

Line 48: are >> were

Lines 56-60:
It has been proven that [the] calprotectin blood level correlates with the degree of adiposity in both healthy children and children with excess body weight [13,14]. It has also been shown that [in adults] calprotectin may be more useful for both detecting and monitoring inflammation [in adults], related to [obesity] metabolic complications [of obesity], as compared to [with]conventional inflammatory markers, such as WBC count or CRP [14,15].

Line 64: the expression genes >>the expression of genes

Line 66: on predisposition >>on the predisposition

Line 74-75: Apart from its cognitive aspect >> This article is far from being Sudoku or cross-words - Please rephrase.

Page 2-3:

Line 86-92: The criteria for inclusion in the study group comprised: body mass index (BMI) z-score ≥ 3 SD in children up to 5 years old [and], BMI z-score ≥ 2 SD in children over 5 years old [it was] as the [diagnostic] criterion for obesity [diagnosis], prepubertal age (first stage of puberty according to Tanner criteria [REF 21 Tanner]), lack of metabolic complications of obesity, [as] established based on physical examination and the results of biochemical tests. The criteria for inclusion in the control group comprised: body weight BMI z- score of < -1 and +1 > SD, prepubertal age, lack of abnormalities in the physical examination and correct results of biochemical tests [20,21].

Please also change: Ref 20 ONIS growth chartsàplease place next to the source cited. The term biochemical test is to vague -be more specific, same with “correct results” - also this is not scientific lingo. Maybe you could put the inclusion/exclusion information in a table to improve readability. Information on inclusion and exclusion criteria is given in Table xx. – the latter just being a suggestion.

Page 4

Line 160: in both studied group >> groups.

Page 6:

Line 176: Interest in answering the question whether obesity in prepubertal children, similarly to obesity in adults and adolescents, is accompanied by chronic low-grade inflammation called metabolic or subclinical inflammation was dictated by the fact that there is little research on this issue in children 178 with obesity of this age [24].

Well now that you have integrated the current literature in your revision, as asked – it seems you would have to revise the above passage. What is the role of the cited reference in this respect?

Line 216: was showed in the study >> was shown in the

Line 220: In contrast to previous observations, we have not demonstrated a relationship between calprotectin and the remaining inflammatory markers.
This remains uncommented. Please add a short discussion about possible reasons.

Line 222ff: The chemerin concentration correlates with BMI as well as adipocyte volume and number and its expression depends on changes in body weight {[37-39].

Remove additional brace, and please revise the syntax of the sentence.

Page 7

Line 224: in obese group >>in the obese group

Line 226ff: In vitro and in vivo studies >> In both in vitro and in vivo [...]

and [an] inverse correlation with anti-inflammatory cytokine – adiponectin was [were] demonstrated

Line 246: Other inflammatory markers, such as ???, >> please name a few with REF

Line 249: It is possible that the use of a more sensitive method for determining IL-6 249 would allow for different, more precise results.
This reviewer somewhat disagrees – maybe it is not the sensitivity so much, but the fact that IL-6 is an acute marker of inflammation and might not be a good marker for “chronic” obesity in children.

Line 259ff:

In the study the potential modifying effect of various external factors on the evaluated inflammatory marker concentrations was also reduced. Due to the fact that some of the studies suggest an effect of physical activity on calprotectin concentrations as well as a dependence of chemerin on the circadian rhythm, all blood tests were taken under the same conditions (i.e., after a night's rest in the morning on an empty stomach).

>>REPHRASE: suggestion

Our study aimed to minimize the effect of physical activity and circadian rhythm as potential external confounder [insert References here] on the concentration of inflammatory markers by reducing preanalytic variance (i.e., overnight fasting, morning blood draws).

Reviewer 3 Report

The authors did a great job on the article and made all the edits. I recommend accepting for publication.
